# The effects of climate change on mental health and psychological well-being: Impacts and priority actions

Shazia Soomro[1] [iD], Dianen Zhou[1] and Iftikhar Ahmed Charan[2]

[1]Department of Sociology, School of Sociology and Political Science of Anhui University, No: 111 Jiulong Road, Jingkai District, Hefei City, Anhui Province 230601, P.R. China and [2]Department of Anthropology, School of Sociology and Political Science of Anhui University, No: 111 Jiulong Road, Jingkai District, Hefei City, Anhui Province 230601, P.R. China

## Research Article

**Keywords:**
climate change anxiety; distress; children; mental health; Pakistan

**Corresponding author:**
Shazia Soomro;
Email: shaziasoomro89@gmail.com

## Abstract

Climate anxiety has a negative impact on the mental health and psychological well-being of the vulnerable population. The goal is to assess many factors that affect mental health and psychological well-being, as well as how climate change affects mental health in Pakistan's vulnerable population. This study provides evidence-based insights into the long- and medium-term impacts of extreme weather events on mental health. To obtain information on these variables, this research uses a quantitative approach and a cross-sectional survey design with a multivariate regression model for empirical tests on a sample of parents and children with an impact on mental health from climate change anxiety. Results indicate that individuals who experience shock climate change anxiety and its effects on mental health and psychological well-being. Climate change can have detrimental effects on children's mental health. (1) Children's Stress Index (CSI): (2) climate change anxiety (CCA), (3) generalised anxiety disorder (GAD) and (4) major depression disorder (MDD), as reported by the children with mental health outcomes. The findings of this study show that climate change has a stressful effect on mental health. The article concludes with a discussion on strategies to address the anticipated mental health issues among children due to climate change.

## Impact Statement

All children experience the psychological impacts of climate change on their mental health, making it a globally significant public health issue. Pakistan is still an underexplored country in terms of climate change's psychological impacts on children's mental health. Pakistan is a low-income country that has a high burden of mental disorders and poor mental health. The present study measure of climate change anxiety demonstrated severe effects on the mental health and psychological well-being of marginalised societies in Pakistan. In Pakistan, people are facing challenges such as climate change, anxiety and mental health. The increasing attention to climate change caused anxiety, which in turn affected the mental and emotional health of children. In Pakistan, several catastrophes, such as extreme weather events and climate change, have increased climate anxiety among individuals. Emotional and cognitive functional impairment factors. Climate anxiety, but not behavioural engagement, was correlated with a general measure of generalised disorder; major depression disorders, stress and depression among younger adults showed higher levels of climate anxiety. There is increasing attention to the negative emotional response associated with climate change awareness.

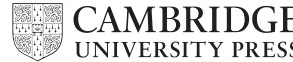

## Introduction

Global societies rightly regard climate change and mental health as two of their greatest challenges. However, insufficient attention has been given to the interaction between and common causes of these two crises. That "the climate crisis is a health crisis" is recognised by global health leaders, including the World Health Organisation (World Health Day, 2022), and recent intergovernmental Panel on Climate Change (IPCC) reports (Lawrance et al., 2022) have outlined that increased global warming will have catastrophic consequences for human health. While the ways climate change negatively affects physical health have been recognised for some time, the effects on mental health have been less well documented, with relevant reviews and policy briefing reports only increasing in the last couple of years (Lawrance et al., 2021; Liu et al., 2020). Mental health refers not just to mental illness, mental problems and mental disorders but also includes states of mental wellness, emotional resilience and psychological well-being (NHS Digital, 2017; Trautmann et al., 2016). Psychological well-being is the interplay between social and psychological conditions that shape human welfare, a broad term that encompasses the states

of being mentally healthy, experiencing mental problems and experiencing mental illness (James et al., 2018; Lawrance et al., 2021).

An increasing body of research suggests the effects of climate change on human health, particularly mental health (Doherty & Clayton, 2011; Hayes et al., 2018). Numerous studies have examined the effects of climate change on mental health, including the potential for linked disasters. According to these studies, there is a higher prevalence of common mental disorders such as generalised anxiety disorder (GAD) and major depressive disorder (MDD). Furthermore, a growing body of research indicates that rising temperatures may exacerbate mental health issues and raise the likelihood of suicidal thoughts and actions (Schwartz et al., 2023). Children in Pakistan and around the world are already being negatively impacted by climate change, which is predicted to worsen in the future. Rising temperatures, severe storms, heat waves, floods, disruption of agricultural cycles, drought and deteriorating water and air quality are just a few of the environmental concerns Pakistan is facing today. These problems have obvious socioeconomic consequences and significantly impact people's quality of life. Because they are still growing physically and intellectually and are more prone to sickness and environmental stressors, children are especially exposed to the harmful effects of climate change (Pandey, 2021). Children are particularly vulnerable to heat waves, and food shortages brought on by droughts can have a negative impact on children's nutrition and health. Young children may find extreme weather frightening and upsetting, which may have an impact on their emotional and psychological health. More young children would be unable to go outside and play as a result of the prolonged length of heat waves, increasing their likelihood of becoming obese (Bhamani et al., 2023; Thiery et al., 2021).

Previous studies have shown that there is a link between climate change and psychological distress. This research suggests that the effects of climate change can be seen in various aspects of a person's mental health. Children are disproportionately affected by the direct impacts of climate change during a period when they are growing physically, psychologically, socially and neurologically. Recent studies have shown that young children are most affected by the indirect consequences of climate change, such as environmental anxiety, which can harm social and psychological health and well-being and may exacerbate the mental health problems of children who already have these problems (Hickman et al., 2021). We are already seeing the effects of wildfires, storms, floods, heat waves and droughts. The slow changes in average temperature, sea level and precipitation patterns that determine the climate in the coming decades may not be as obvious, but ultimately, they are more important because they will harm more people. Despite the perception of polar bears as the primary victims of climate change in several cases (Clayton, 2020), concern for human well-being is becoming increasingly evident. In addition to the severe impact of natural disasters and the social effects of forced migration and conflict, increased heat and vector-borne diseases, as well as malnutrition, will threaten physical and mental health (Born, 2019; Clayton, 2020; Watts et al., 2019).

It is also important to consider that a rapidly growing body of research has identified links between climate change and a variety of negative impacts on children's mental health outcomes, particularly for children and youths (Trott, 2022). There are several possible explanations for the relationship between the above-mentioned health outcomes. Researchers and experts in climate change and environmental education are increasingly aware of the

potential of their potential to include more than just skillful information, for example (affective and attitudinal) children responses to action climate change outcomes psychological well-being (Chawla, 2020; Ojala et al., 2021). To develop climate change education strategies that promote children's overall well-being and constructivist climate change participation. Instead, it is a reason to steer clear of this subject in class and at school events. It is imperative to comprehend the entire spectrum of children's psychological experiences about climate change (Halstead et al., 2021; Trott, 2019, 2020). Furthermore, investigating the attitudinal and affective aspects of children's involvement in climate change may shed light on the potential context-specific effects of climate change education (Trott, 2022).

This study investigates the present state of evidence and knowledge about the impacts of climate change on mental health. This research pays particular attention to the inequitable impacts of climate change on the mental health of marginalised and vulnerable populations. Despite evidence of the high prevalence of children's stress among vulnerable children in Pakistan, it is unclear to what extent this stress is related to variables such as children's perceptions of generalised anxiety disorder (GAD) and MDD. In this study, we examine the most recent findings about climate change's impact on children's mental health in relation to both direct and indirect effects. It is crucial to consider how children learn about climate change and how they cope with climate anxiety, such as fear, phobia and stress related to it, for their mental health. The climate crisis and climate anxiety have triggered existential concerns in both parents and their children. While climate change anxiety (CCA) is gaining attention in the popular press (Benoit et al., 2022), not much study has looked at the relationship between CCA and mental health symptoms, such as GAD and MDD symptoms. Furthermore, since more people—especially young people—are taking action to combat climate change, additional study is required to determine the potential relationships between action and symptoms of mental health, as well as anxiety related to climate change (Schwartz et al., 2023).

The primary aims of this study are to examine the notable impacts on physical health, mental health and psychological well-being caused by exposure to extreme weather events associated with climate change, as extensively recorded in previous research. To assess Pakistan's climate resilience, this research presents a novel method for evaluating the consequences of climate change on children's mental health. The study also examines the less well-documented effects of climate change on children's physical health, as well as how it affects their psychological and mental well-being. To investigate how mental health may be affected by climate change in the present and the future through emotional reactions like elevated worry. We aim to understand the factors that trigger children's anxiety about the climate and its impact on their psychological and mental well-being. To discuss the most recent research on the psychological implications of climate change on kids and the direct and indirect effects it has on mental and physical health.

## Literature review

### Climate change education and action

Both before and after independence, Pakistan placed a high value on education and literacy, but they were unable to move beyond rhetoric and become tangible. Experts believe that the general public's comprehension of environmental literacy and climate

change education is remarkably low. Postcolonial countries have not made significant efforts to adapt their educational systems to meet regional or national needs. Although Pakistani high institutions provide the degree courses associated with environmental studies, they have been introduced for postgraduate and graduate programme students. In the process of primary to high school education (grades 1–10), children still face a lack of environmental textbooks to develop awareness and knowledge at an early stage in their lifetime (Iqbal & Khan, 2020). According to a review of Pakistan's elementary school curricula, textbooks issued by the government for social study or general science subjects contain chapters on basic environmental phenomena. The study revealed that teachers often completely ignore these chapters in classroom curricula, giving only secondary attention to theories based on physics and chemistry. This practice method provides insights into educators' priorities. The study examined existing policies and children's attitudes towards environmental and climate change among the public and private schools' textbooks of science, geography and social studies for grades 5–8 and 9–10. It has been reported that these textbooks lack knowledge and awareness about the environment and climate change (Shaw, 2015).

To address the relevant issues of climate change and environmental degradation, educational policymakers came to the conclusion that there is no effective framework in place, let alone one that would provide background and policies for climatic adaptation and mitigation at different levels. Pakistan's educational system was decentralised to answer the 18th Amendment, and the authority to create and execute education literacy curricula was shifted to the provinces. To provide knowledge of climate change and environmental challenges to pupils as well as a framework that fosters innovation and critical thinking that develops a climate-positive attitude, provinces should have taken advantage of this fantastic opportunity to customise their regional curricula (Iqbal & Khan, 2020). Climate change education, a field in its infancy, explains the importance of incorporating it into school curricula worldwide, especially in underdeveloped nations like Pakistan. Despite the implementation of certain measures, policymakers and relevant policies seem indifferent to the potential for enhancing processes that could aid in climate change adaptation and mitigation. This is despite the importance and necessity of integrating climate change into Pakistan's basic education. In addition, there is a need to develop policy guidelines for integrating climate change adaptation and mitigation strategies into elementary education curricula through coordinated education and environmental governance mechanisms at the federal and provincial levels.

### Psychological trauma and mental health in the context of climate change

The evidence of the impacts of climate change on mental health and well-being is growing rapidly. This study aims to determine the extent and characteristics of existing interventions targeting psychological trauma and mental health issues related to climate change's psychological and mental health consequences. Public health emergencies related to climate change are becoming more widely acknowledged. In addition to its widely recognised effects on physical health, the crisis has a significant influence on mental health (Xue et al., 2024). Researchers have linked climate events to exacerbated psychiatric mortality outcomes, such as depression, posttraumatic stress disorder (PTSD) and suicide (Charlson et al., 2021). Most population groups, including children and teens, elderly individuals, pregnant women, those with chronic conditions

and marginalised populations, have been recognised as being particularly at-risk (Charlson et al., 2021; Cianconi et al., 2020). Indigenous communities around the world have spoken of sentiments of despair, rage, grief, fear and powerlessness brought on by forced migration, worry due to climate change, broken cultural continuity and past and present disempowerment (Middleton et al., 2020).

There is a growing body of research on the psychological response and coping mechanisms associated with climate change, but little is known about evidence-based interventions that can lessen the negative effects and help individuals and communities. A scoping review on the connection between climate change and mental health research identified 120 original studies, the vast majority of which were cross-sectional studies that examined how exposure to climate change affected mental health outcomes (Charlson et al., 2021). The eight studies that dealt with interventions were mainly theoretical and lacked strong data. Only two studies in another assessment of eco-anxiety therapies had an empirical evaluation component; the remaining studies were conceptual reflection articles (Baudon & Jachens, 2021). The most thorough evaluation to date found 23 studies (Palinkas et al., 2020), but it did not distinguish between interventions used for climate-related incidents and those for other contexts (like armed conflicts) that are theoretically relevant to the context of climate change. The review was also restricted to scholarly works and activities aimed at treating or avoiding recognised mental illnesses. According to the World Health Organisation (WHO), mental health is a general state of well-being rather than only the absence of a condition (Xue et al., 2024). In the face of climate stress, interventions that foster psychological strengths and emotional resilience may be easier to identify with a more comprehensive conceptual framework.

### Cognitive CCA and mental health

Climate change information has a variety of effects on individuals' psychological views, whereas different types of climate change events are associated with distinct patterns of emotional regulation. Occasionally, people may feel distressed due to climate change without realising it, merely feeling a general sense of unease with the ongoing environmental changes (Cianconi et al., 2021). Multiple severe weather events in one's area can elicit a stronger response because people perceive the danger as tangible and nearby (Scheffer, 2010), whereas direct observations of climate change often perceive it as more abstract and distant (Akerlof et al., 2015). Consequently, a decrease in "psychological distress" correlates with an increased level of concern and stronger intentions to act in an environmentally friendly manner, leading to improved adaptations. Climate change's effects on mental health can be categorised as acute or chronic; however, some effects may overlap. They can vary from mild stress to serious problems. Acute effects typically result from severe and intense weather events or natural disasters, such as hurricanes, floods, wildfires, tornadoes and droughts. These events often come suddenly and without notice, leading to the loss of lives, resources, social support and social networks. Generally, we identify these symptoms as acute stress disorders if they manifest within 4 weeks of a tragedy. A longer period of persistence may lead to a diagnosis of PTSD, anxiety or depression (Cianconi et al., 2021).

Arguably, cognitive evolution should have favoured processes that foster not only individual but also group effectiveness. For example, collective functioning is crucial when it comes to hunting

big game. As a result, cognitive heuristics should be able to integrate information about the intraindividual and individual–environment relationship spheres. Climate change's impacts on adults' physical and mental health have been significant and wide-ranging. There is smaller but growing research on how climate change affects children and youth psychologically. It is still very modest, but it is rising. Not every child will experience the same impact. Those living in areas that are most vulnerable to climate change, with poor infrastructure and fewer supports and services, are most likely to be affected (Burke et al., 2018). The study analysed how Pakistan's educational outcomes were severely affected by extreme weather events. How children's mental health and anxiety affect climate change. Every year, increasing climate change risks are demonstrated through powerful extreme weather events, including heat waves, heavy rainfall and droughts that severely affect children and people. The likelihood and severity of occurrences are increasing, which has detrimental repercussions for people, property and the environment. Extreme weather events are distinct from other climate change consequences in that they are noticeable immediately and are poorly defined by the climatological methods investigated in many projections (Clarke et al., 2022). Unfortunately, there are no standardised methods or efforts to systematically document the hazards associated with climate change. Severe monsoon weather has affected Pakistan since mid-June 2022. Since then, the situation has deteriorated significantly, as the rainfall has been equivalent to nearly three times the national 30-year average.

In recent years, rapidly growing research has revealed a connection between climate change and schooling, as well as a number of psychological effects on children and adolescents. Floods are the main consequence of extreme rainfall. Changes in flood risk caused by heavy precipitation differ in other factors, including language changes, river management and regional sensitivity to floods (Ji et al., 2020); furthermore, some factors are climate-related, such as snowmelt, soil moisture and storm size (Clarke et al., 2022). Although there are large regional and subregional variations in river flow trends, many of the observed changes can only be attributed to anthropogenic climate change. The consequences of climate change precipitation are increasing flood frequency and severity, according to evidence from the literature on attribution science. In August 2022, unusually strong monsoon rains triggered one of the worst floods in Pakistan's history, affecting approximately 33 million people nationwide (Qamer et al., 2023). As a result, 85% of children living in developing countries, as well as a small number of disadvantaged children in developed countries, will be the most severely affected.

### CCA: Direct impact on children's mental health

The main impact of climate change on children's mental health and psychological well-being is anticipated to be the rise in severe extreme weather events. The impact of such traumatic events on children is extensively documented in the literature. Children are particularly vulnerable to the effects of extreme weather, including disasters that cause family stress, damage to social support systems and the displacement of homes and communities. There is a significant risk of developing additional mental health issues, such as attachment disorders, phobias, anxiety, panic attacks, and depression (Burke et al., 2018). There are numerous scientific indications suggesting that human-induced climate change is already having adverse effects on the ecosystem and jeopardising the world's food security. The recent increase in the frequency of floods and other extreme events throughout South Asia and the rest

of the world, particularly in Pakistan, has resulted in significant losses and destruction.

In addition to the effects of EWEs on mental health problems, other psychological effects of traumatic experiences in climate-related anxiety and their disruptions can include negative impacts on children's capacity to regulate emotion, increased cognitive deficits, behavioural problems, learning problems, mental development problems, impaired language development, adjustment problems and an understanding of academic performance (Burke et al., 2018). The EWE and disaster impacts on children's capacity to perform academic education range from whole destruction to negative impacts on children's mental health performance (Gibbs et al., 2019; Mudavanhu, 2014). The available statistical data indicate that heavy rainfall events have severely damaged several school buildings due to disasters (Amri et al., 2022). It is clear that for approximately 1.6 billion students worldwide, unexpected and sudden transitions from physical activities in the classroom to online learning present a significant barrier. Thus, it is important to comprehend how natural disasters affect students' mental health, psychological well-being and safety, as well as their ability to study and how any possible losses in learning quality can be identified and efficiently addressed.

### CCA and activism among children

Children experience more climate anxiety as a result of climate change, and these feelings may include anger, fear, sadness, despair, concern, guilt, shame and hope, although their presence varies between individuals. Certain emotions, particularly grief, anxiety and worry related to recent and upcoming losses, have gained increased attention as the area of study develops. Research on other emotions, such as how people feel guilty about their contributions to climate change or ashamed of the greater climatic damage caused by humans, has only recently started (Hickman et al., 2021), among participants, the majority of children's perceptions of climate-related anxiety significantly have negative effects on mental health and psychological well-being. These studies have experimented with a variety of pedagogical approaches in an effort to address and examine the psychological experiences of a wider range of learners in relation to climate change and climate-related anxiety to increase the informational potential of climate change education. For instance, the political and justice aspects of climate change have received more focus in climate change education (Jorgenson et al., 2019; Stapleton, 2019; Waldron et al., 2019), however, integrating creative expression and arts into children's attitudes towards climate change education and climate-related mental health and anxiety.

According to this research, allowing for critical reflection, collaborative meaning-making, agentic action and creative experimentation can lead to more transformational outcomes than methods that only emphasise knowledge (Trott, 2022; Trott et al., 2020a, 2020b). However, most current methods of educating people about climate change still emphasise how to teach young children and adults the scientific aspects of the problems facing our society rather than the social or traditional aspects and how to solve these problems. Scientific and technological solutions, rather than political or cultural approaches, are emphasized. Additionally, current research on climate change education involves students participating in real-world climate change activities connected to their studies. Inquiring into the transforming effects of climate change education to reframe students' perceptions or engaging in cooperative, community-based climate change activities in particular (Monroe et al., 2019;

Stapleton, 2019). Incorporating climate change adaptation strategies into primary education curricula and providing environmentally friendly information and scientific knowledge to promote a climate-compatible development agenda is one of the challenges of environmental governance. By raising children's and young people's understanding of pertinent actions, triple-win solutions can be used to promote development that is compatible with climate change. Particularly in poor countries, basic education has yet to address the most serious problems of climate change by developing environmental knowledge, transmitting the necessary skills and fostering emotional attitudes at a young age.

## Methodology

To gather descriptive data for their research, researchers frequently employ questionnaires, observations and interviews, in addition to applying relevant literature and creating well-structured questionnaires (Gay et al., 2012). This research was founded on a survey. To extrapolate to the full population, survey research examines the relationships between variables and population subsects (Bano et al., 2022; Pinsonneault & Kraemer, 1993). This study investigated the relationship between children's perceptions of the impact of climate change on mental health. Drawing on the children's adjustment and adaptation response theoretical model, this study examined climate change education, action and stress levels in children, encompassing various factors that may contribute to overall stress. This study uses CSI and mental health as measures for cognitive and emotional impairment, functional impairment, CCA, GAD and MDD. We hypothesise that both parental perceptions of children's vulnerability to climate anxiety protection would be significantly related to children's mental health, beyond the impact of children's characteristics. We used a cross-sectional survey design with a sample of parents and children with an impact on mental health and climate anxiety to obtain information on these variables. As previously mentioned, the consensus quantitative research approach developed by Hill et al. (2005) is centred on the varying perspectives and experiences of the research team members as well as the inferences they can make from their assumptions. It was ensured that the ethnic composition of the sample was distributed equally. Snowball sampling was applied to a few events (Bhamani et al., 2023).

### Study setting, sampling and participants

According to Mendenhall, a population is a collection of people that a researcher wants to learn more about (Black & Mendenhall, 1990). Pakistani parents of children with mental health who met the following criteria were recruited between December 2022 and March 2023 from multiple trauma release and wellness centres, Karachi Psychiatric Hospital organisations and the Education Department of Sindh. Participants were included if they were (a) children diagnosed with mental health; (b) their children suffering from anxiety, depression, fear, stress or mental illness were diagnosed before age 16; and (c) it had been less than 10 years since their children received trauma treatment. We obtained data from 6- to 16-year-old children from the first year of government high school and their parents. Thus, respondents' parents and children (N = 163, 117 children and 46 parents) were included in the analyses. The children were selected from all types of secondary and high schools in different regions of Sindh province because these regions are among the most flood-affected in Sindh, like Dadu, Jacobabad, Kambar, Shahad Kot and Khairpur. Furthermore, we conducted this study in different areas of ongoing working schools. We informed the study's participants about their voluntary participation, the confidentiality of the data they submitted and their right to withdraw from the study at any time. We used the questionnaires to collect sociodemographic background information about the children, such as their names, ages, educational levels and duration of the study.

### Procedures

Participants were recruited using multiple strategies to ensure effective access for eligible individuals residing and studying in places such as schools in remote and rural areas in Sindh. Initially, the research team visited two major trauma release and wellness centres, Karachi Psychiatric Hospital and health organisations within the community. Additionally, participants were recruited at children's climate-related anxiety events, such as parental education sessions held throughout Sindh. Second, we engaged with stakeholders and the District Health Organisation (DHO) to collaborate with support groups for parents of children undergoing trauma treatment in hospitals. We sent a brief letter about the study to the group members to recruit participants. Additionally, the team conducted interviews with children who had recently experienced a full range of mental health outcomes impacted by climate change.

Finally, when eligible participants approached our research team, we provided detailed explanations of the study's aims and objectives, as well as any potential risks and benefits involved. We emphasised the voluntary nature of participation, assured confidentiality and provided incentives for participation where applicable. We obtained written informed consent from each participant, adhering to the regulations of the university's ethics committee, where the principal investigators are affiliated.

### Measures

**The CCA scale** (Clayton & Karazsia, 2020) was used to measure CCA. The 13-item scale in the measure evaluates the detrimental effects on cognition and emotion as well as functional impairment associated with understanding climate change. The assessment consists of two subscales: one that evaluates cognitive-emotional impairment with eight items and another that measures functional impairment with five items. Items on the subscale consist of cognitive-emotional impairment "I go away by myself and think about why I feel this way about climate change," but factors like these are included in the Functional Impairment subscale. "My anxiety about climate change interferes with my ability to get work or academic performance done." We used a Likert scale to report each issue, with response possibilities ranging from 1 (strongly disagree) to 5 (strongly agree). We calculated the mean scores, which indicated higher levels of climate anxiety. In a sample of children and their parents from Pakistan, this scale demonstrated strong reliability and validity (Clayton & Karazsia, 2020; Reyes et al., 2021; Schwartz et al., 2023). The current sample's reliability was deemed acceptable score ( $\alpha = 0.90$; subscales measuring cognitive and emotional impairment $\alpha = 0.85$; and functional impairment $\alpha = 0.93$).

The Patient Health Questionnaires GAD Symptoms Scale was used to examine the symptoms of GAD (Spitzer et al., 2006). This GAD was utilised in the Fourth Edition of the Diagnostic and

Statistical Manual of Mental Disorders (DSM-IV: American Psychiatric Association & Association, 1994). The four symptoms were described, and participants were asked to indicate how often each one had afflicted them over the past 3 weeks. Response options ranged from 1 (definitely disagree) to 3 (definitely agree). The GAD has adequate internal reliability, high-retest reliability for GAD symptoms and scores equal to or greater than 10 indicative of probable GAD. Reliability in the present sample was the total score ($\alpha = 0.79$).

**MDD symptoms:** Utilising the eight-item Patient Health Questionnaires, symptoms of MDD were assessed (Kroenke et al., 2009). The Diagnostic and Statistical Manual for Mental Disorders, Fourth Edition (DSM-IV), which excludes symptoms of suicidality, has four subscales that have criteria for MDD that are assessed using the 25 items in PHQ-8 (DSM-IV: American Psychiatric Association & Association, 1994). We asked the participants to report the four symptoms and the frequency of their discomfort during the previous 3 weeks. Each item has a 4-point scale: 0 = never, 1 = sometimes, 2 = most of the time and 3 = always. We calculated sum scores, where higher scores indicate more severe MDD symptoms and scores of 10 or higher indicate a likely diagnosis of MDD. In the current sample, reliability was the overall score ($\alpha = 0.84$).

### Statistical analysis

We performed the regression and correlation analyses using IBM SPSS 26, a popular statistical software programme. IBM SPSS 26 is intuitive software with a variety of essential features that allow researchers to investigate the relationship between variables and the predictive power of independent variables on a dependent variable. Moreover, SPSS 26 was instrumental in determining the strength of the association and the nature of the relationship between variables by conducting correlation analysis. The investigation of potential connections between the different variables was done using the Pearson product–moment correlation (Kurtz & Mayo, 1978). We used regression analysis to identify the significant factors. A study can extract a smaller set of variables from a large number of predictors by using multiple regression analysis, which streamlines the data, boosts predictable accuracy and decreases the number of unnecessary predictors (Halinski & Feldt, 1970). The questionnaire's dependability serves as a gauge for the consistency of the computation method used to get the results. The survey results, as well as the conclusions drawn from the five tested hypotheses, are listed below.

Three steps made up the quantitative analysis that was carried out using IBM SPSS 26. We first conducted a series of exploratory analyses, which involved creating a correlation matrix and calculating descriptive statistics for each variable in the investigation. Second, descriptive analyses were conducted to understand the levels of the Children's Stress Index (CSI), climate change anxiety (CCA), GAD and MDD. To examine the main research questions, three multiple regression analyses were conducted to assess how CCA may be related to perceived GAD and MDD, adjusting for direct climate change experience and demographic factors. In the three multivariable models, the dependent variable climate change anxiety was measured against three CSI subscales: CCA, GAD and MDD. We reverse-coded several CSI items and used the sum of 12 items under each subscale to calculate the three CCA subscale scores. Finally, CSI total scores were calculated by summing the three subscales' scores. For the independent variables, the CCA, GAD and MDD total scores were calculated as the sum of the item scores as a continuous variable.

## Results

### Survey results

Table 1 represents the characteristics of the collected sample and reveals that the mean age of the children was 11.5 with an SD of 4.87. More boys counted: 68, accumulating 42% of the total sample, and the girls were 49, making 30.2% of the sample. The majority of the children and their parents were from rural areas, making up 94.5% of the total, and urban area parents and their children were only 5.5% of the sample. Children in their high schools were 88, making up 74.8% of the sample, and those who were in secondary school were only 29, making up 25.2% of the total sample. There were 86 children with cognitive-emotional impairment, making up 73.6% of the sample, while functional impairment children were 11, which was 9.2% of the total. Children with CCA were 20 and made up 17.2% of the total sample. While interviewing the parents, it came to our knowledge that for those children who were diagnosed with GAD symptoms, the mean age was 8.1 with an SD 4.34, and for those who were diagnosed with MDD symptoms, the mean age was 12.16 with an SD 1.45. Some parents who knew their children were diagnosed with either stress

**Table 1.** Descriptive statistics of the sample

| Variables | M (SD) | n (%) |
|---|---|---|
| Ages | | |
| Parents | 36.4 (3.85) | |
| Children | 11.5 (4.87) | |
| Gender identity | | |
| Male | | 29 (17.9%) |
| Female | | 16 (9.9%) |
| Boys | | 68 (42%) |
| Girls | | 49 (30.2%) |
| Residence areas | | |
| Urban | | 9 (5.5%) |
| Rural | | 154 (94.5%) |
| Student status | | |
| Secondary school | | 29 (25.2%) |
| High school | | 88 (74.8%) |
| Children's Stress Index (CSI) | | |
| Cognitive-emotional impairment | | 86 (73.6%) |
| Functional impairment | | 11 (9.2%) |
| Climate change anxiety | | 20 (17.2%) |
| Mental health | | |
| GAD symptom diagnosis | 8.10 (4.34) | |
| MDD symptoms diagnosis | 12.16 (1.45) | |
| Children diagnosis | | |
| Yes | | 42 (92.0%) |
| No | | 4 (8.0%) |
| Trauma treatment | | |
| Yes | | 24 (51.2%) |
| No | | 22 (48.8%) |

**Table 2.** Descriptive statistics of Children's Stress Index: CCA, GAD and MDD scales

| Variable | $N$ | Possible Range | Mean | SD | Min | Max | Risk group (%) |
|---|---|---|---|---|---|---|---|
| Children's stress index | 117 | 36–180 | 86 | 19.19 | 43 | 139 | 15.7 |
| Climate change anxiety | 117 | 12–60 | 32.69 | 8.42 | 12 | 56 | |
| Climate change education and action | 117 | 12–60 | 25.38 | 7.64 | 13 | 47 | |
| Mental health | 117 | 12–60 | 28.38 | 8.43 | 13 | 56 | |
| Generalised anxiety disorder | 117 | 0–24 | 11.03 | 4.2 | 0 | 21 | 68.8 |
| Major depression disorder | 117 | 0–75 | 31.86 | 9.49 | 9 | 63 | 14.4 |

CCA = climate change anxiety; GAD = generalised anxiety disorder; MDD = major depression disorder.

level or mental health were 92%, and those who did not know were only 8%. Similarly, for the treatment of their children, those parents who had any trauma treatment for their child were 51.2%, and those who had never had it were 48.8%.

Table 2. Children's Stress Index (CSI): climate change anxiety (CCA), generalised anxiety disorder (GAD) and major depression disorder (MDD) as reported by the children with mental health outcomes. The mean CCA total score was 86 (SD = 19.9) with a range of 43 to 139, and approximately 15.7% ($n = 18$) of the children experienced a clinically significant level of stress (>90th percentile). The mean score on the GAD was 11.03 (SD = 4.20) and 68.8% ($n = 80$), scored 10 or greater, implicating that they perceived children as highly vulnerable. The mean score on the MDD was 31.86 (SD = 9.49) and 14.4% ($n = 17$), of the children reported climate change education-protecting behaviour towards their mental health outcomes.

Correlation analysis between study variables is shown in Table 3. It was shown that the Children's Stress Index (CSI) was found to be intercorrelated. CCA was positively correlated with ICA ($r = 0.524, p < 0.01$) and MH ($r = 0.513, p < 0.01$); ICA was positively correlated with MH ($r = 0.677, p < 0.01$). Perceived GAD was correlated with the three subscales of the CSI ($r = 0.415, p < 0.01; r = 0.247, p < 0.01; r = 0.287, < 0.01$), for CCA, ICA and MH, respectively. MDD was correlated with CCA ($r = 0.511, p < 0.01$) and MH ($r = 0.303, p < 0.01$). A moderate correlation existed between perceived GAD and MDD ($r = 0.369, p < 0.01$).

Table 4 shows the results of multiple regression analyses. The first model accounted for 32.2% of the variance in CSI scores ($F(1.143) = 7.648, p < 0.001$). Higher levels of CCA ($\beta = 0.245, p < 0.01$) and mental health ($\beta = 0.410, p < 0.01$) were associated with higher CCA scores. The second model accounted for 7.6%

of the variance in climate change education and action CCEA scores ($F(1.143) = 2.158, p < 0.05$). Higher levels of CCA ($\beta = 0.219, p < 0.05$) were associated with higher ICA scores, but not mental health. The third model accounted for 10.6% of the variance in mental health scores ($F (1.143) = 2.668, p < 0.01$). Higher levels of CCA ($\beta = 0.277, p < 0.01$) and mental health ($\beta = 0.178, p < 0.05$) were associated with higher ICA scores.

## Discussion

Although long-term effects are more difficult, reactions to extreme weather events are comparable to the trauma caused by natural catastrophes. Climate change pressures could result in climate-related mental diseases (CRMD), new adoptions and new types of mental health and misery, all of which could result from the pressures of climate change (Cianconi et al., 2021). This study examined three children's capacity variables: CCA, GAD and MDD among the children in Pakistan. Specifically, this study sought to investigate whether children's perceptions of CCA, GAD and MDD were related to their stress after controlling for their characteristics. We found that a significant number of children experienced a clinical level of stress (Charlson et al., 2022). The study includes a relatively high impact on children's mental health by climate change in Pakistan, including a high number from both rural and urban regions from different schoolchildren. Climate change represents the biggest psychological, physical and mental health threat of the 21st century. We urgently need to enhance our comprehension of the impact of climate change on mental health and devise strategies to prevent mental health issues.

How to advocate for mental health through clinical work at the levels of children, parents, communities and society in the context of climate change. Most qualitative researchers take it for granted that open-ended questions are more likely to provide richer data (or induce more storytelling). This study examined the relationships between CCA and mental health concerns, schoolchildren's actions and children's attitudes towards perceptions about climate change's psychological effects on children's mental health. Open-versus-closed questions and focus groups in the literature provided a more thorough explanation of children's perspectives during interviews. Recent indications suggest that the negative feelings of poor and marginalised communities, both directly and indirectly affected by climate change, as well as other distressing emotions and thoughts about climate change, counteract children's feelings of concern and impact their daily lives. The results of this study imply that mental health, GAD and MDD in Pakistan are negatively correlated with children who experience emerging CCA. The implications of climate change on children's mental health and well-being are consistent with the need for thought and action. The

**Table 3.** Correlation analysis of CCA, CCEA, MH, GAD and MDD

| | 1 | 2 | 3 | 4 |
|---|---|---|---|---|
| Climate change anxiety | | | | |
| Climate change education and action | 0.524* | | | |
| Mental health | 0.513* | 0.677* | | |
| Generalised anxiety disorder | 0.415* | 0.247* | 0.287* | |
| Major depression disorder | 0.511* | 0.156* | 0.303* | 0.369* |

CCA = climate change anxiety; CCEA = climate change education actions; MH = mental health; GAD = generalised anxiety disorder; MDD = major depression disorder.

*$p$ value < 0.05.

**$p$ value < 0.01.

***$p$ value < 0.1.

**Table 4.** Regression analysis of factors associated with CSI, CCA, GAD and MDD

| Variables | Children stress index | | | Climate change anxiety | | | Generalised anxiety disorder | | | Major depression disorder | | |
|---|---|---|---|---|---|---|---|---|---|---|---|---|
| | $b$ (SE) | $B$ | $t$ | $b$ (SE) | $B$ | $t$ | $b$ (SE) | $B$ | $t$ | $b$ (SE) | $B$ | $t$ |
| Age | 0.446 (0.411) | 0.111 | 1.084 | 0.189 (0.163) | 0.107 | 1.131 | 0.281 (0.176) | 0.176 | 1.592 | 0.063 (0.193) | 0.035 | 0.326 |
| Children age | −234 (0.483) | −0.051 | −0.485 | −0.018 (0.196) | −0.009 | −0.092 | −0.013 (0.207) | −0.007 | −0.062 | −0.266 (0.226) | −0.131 | −1.178 |
| Gender identity[a] | 1.089 (2.879) | 0.028 | 0.378 | −072 (1.168) | −0.004 | −0.061 | 0.181 (0.1.234) | 0.12 | 0.147 | 0.832 (1.348) | 0.048 | 0.617 |
| Student status[b] | −3.144 (3.586) | −0.069 | −0.877 | 0.316 (1.455) | 0.016 | 0.217 | −0.861 (1.537) | −0.048 | −0.56 | −2.012 (1.679) | −0.101 | −1.198 |
| Diagnosis | −4.576 (3.327) | −0.104 | −1.375 | 0.206 (1.350) | 0.011 | 0.153 | −3.446 (1.426) | −0.198 | −2.416* | −.748 (1.558) | −0.039 | −0.48 |
| Trauma treatment[d] | −2.793 (2.908) | −0.072 | −0.961 | −0.851 (1.180) | −0.05 | −0.722 | −1.177 (1.247) | −0.077 | −0.944 | −681 (1.361) | −0.04 | −0.5 |
| Climate change anxiety | 1.281 (0.372) | 0.275 | 3.443** | 0.501 (0.151) | 0.245 | 3.319** | 0.0405 (0.159) | 0.219 | 2.539* | 0.466 (0.174) | 0.227 | 2.676** |
| Mental health | 0.537 (0.169) | 0.261 | 3.176** | 0.370 (0.069) | 0.41 | 5.388*** | 0.043 (0.073) | 0.053 | 0.592 | 0.161 (0.079) | 0.178 | 2.036* |
| $F$ | 4.758*** | | | 7.648*** | | | 2.158* | | | 2.668** | | |
| $\triangle R^2$ | 0.268 | | | 0.370 | | | 0.142 | | | 0.170 | | |
| Adjusted $R^2$ | 0.212 | | | 0.322 | | | 0.076 | | | 0.106 | | |

CSI = children stress index; CCA = climate change anxiety; GAD = generalised anxiety disorder; MDD = major depression disorder.
*$p$ value < 0.01.
**$p$ value < 0.05.
***$p$ value < 0.1.

psychological effects of climate change significantly impact children's negative emotions and mental health.

In Pakistan, an underdeveloped country, the majority of children living in the marginalised community face various problems related to education, such as lack of infrastructure, lack of awareness, lack of textbooks, lack of information related to science and social science, lack of good faculty, lack of education for climate change and environmental challenges at all levels of the vulnerable population. Several scholars have highlighted these issues. The vulnerable population, particularly children aged 10–20, suffers from CCA, and children experienced severe floods in the history of Pakistan in 2022. CCA is fundamentally distressing about climate change and its impacts on the landscape and human health. But CCA can also take a toll on our mental health, triggering climate anxiety, or more broadly, eco-anxiety and affecting people who are concerned about the state of the climate and the environment. Therefore, we need to improve climate education for children to better understand the issues impacting their psychological and mental health. However, our study found significant differences in how to cope with climate change and its psychological impacts based on gender, education and socioeconomic level. The girls appeared to be more sensitive to thought and action than the boys; this may be a result of their likely proenvironmental sentiments.

In this study, CCA was negatively associated with the ICA, MH, GAD and MDD subscales. Previous studies have found that increases in CCA may be because girls and boys differ in a variety of ways related to the gender pay gap, including value systems such as altruism and consideration. Additionally, researchers have found that girls worry about certain environmental problems, specifically regional problems that increase the risk of mental health (Bhamani et al., 2023). Effective interaction is essential for making decisions, reflecting on them and enchanting action to solve these issues because climate change can have a significant negative impact on both public health and children's health. The study found that the impact of CCA on children's mental health and a comprehensive understanding of children's perceptions of the latest evidence on their psychological experiences in the context of climate change are critical to the development of approaches that encourage and help children's constructive engagement and, largely, mental health. In addition, parents' awareness of climate change and mitigation is evident in their children, including the fact that family climate action can significantly influence children and entire families to take action.

## Implications and limitations

Several limitations to this study should be acknowledged, despite the fact that it makes significant contributions to the field. Initially, our reliance on cross-sectional data limited us to describing associations rather than directions between the constructs. People who are experiencing mental health difficulties, for example, are likely less able to participate in group activities and more susceptible to the negative impacts of anxiety related to climate change. Furthermore, the instrument utilised to evaluate climate change education and action had various shortcomings. These included the fact that certain items required both emotional and action-related responses and that the measure did not encompass the full range of individual actions that respondents detailed in their open-ended answers. Second, support groups recruited the participants. Therefore, the children in our sample may have had more distress or concern about CCA and mental health compared to those not involved in support groups. Third, children's perceptions of climate anxiety,

climate change education actions and mental health may be different. Further investigation is necessary to develop more comprehensive measures for individual actions aimed at reducing climate change and to ascertain whether broader measures could provide different results. Although the quantitative data alone provided some understanding of the participants' opinions, the qualitative data supplied offered a more comprehensive appreciation. However, conducting in-depth interviews with focus groups might yield even better insights into their experiences of climate change worry. Despite these limitations, this research significantly contributes to understanding the anxiety emerging children experience due to climate change and its correlation with clinical measures of GAD and MDD. Additionally, it emphasises the potential for engaging in community efforts to enhance the welfare of those who are facing anxiety related to climate change.

## Conclusion

There is growing awareness of the detrimental effects of climate change on mental health and psychological well-being. This research has examined the potential causes of climate-related anxiety that have impacted the mental health and psychological well-being of children in Pakistan. To conclude, our findings suggest that climate anxiety not only impacts mental health but is also a psychological problem for young children. Furthermore, climate-related anxiety makes clear the critical importance of children's changing attitudes towards climate change education. Regardless of how children are measured in various environments, such as classroom learning, children's emotional, psychological and mental health realities inevitably play a role outside of academic development. As we interpreted, children's perceptions about climate change are not always 'pessimism' especially when children's knowledge is paired with opportunities for critical dialogue, reflection and action. The findings of this study were a significant influence on climate change and psychological impact on children, both direct and indirect, between urban and rural areas. Effective climate change education programmes have reduced children's climate anxiety. However, we found significant differences in terms of gender, qualifications and socioeconomic status. Finally, researchers, practitioners of mental health and environmental experts have made significant contributions to both mitigating the effects of climate change and developing strategies to support children in adjusting to its effects. It is also recommended to intensify future research on the effects of climate change on the relationship between children's public health and their psychological well-being. The 10- to 16-year-old children in this study gained increased awareness of climate change and its impacts, such as psychological and mental health, among a deeper interest in a sense of respect for the environment and their responsibility concerning it.

Although environmental issues have received more attention in recent years, relatively few young people have acknowledged the importance of climate change education and its effects on mental health. It is time to seriously consider the ways that climate change may harm psychological and mental health in light of increasing data. Environmental experts and psychological professionals should have climate change on their radar. This might mean awareness and education about environmental issues should be part of the school-level training of children, as well as the education of parents and the community through psychology and health clinical training. This may mean focusing on cultivating an

understanding of best practices to help children and adolescents who are experiencing climate change and psychological anxiety reflect and take action.

**Open peer review.** To view the open peer review materials for this article, please visit http://doi.org/10.1017/gmh.2024.65.

**Data availability statement.** The data supporting the findings of this study are available upon reasonable request from the authors.

**Acknowledgements.** We extend our heartfelt gratitude to all communities, medical doctors and school staff who wholeheartedly participated in and supported the implementation of mental disorders competencies in our research project. This research was supported by the Anhui University Collaborative Innovation Project of General History of Taiwan Ethnic Minorities Support Agency under Contact no. GXXT-2021-034.

**Author contribution.** All authors participated in drafting the research design and planning the data collection. Each author made an equal contribution to this work. All authors wrote, reviewed and commented on the manuscript. All authors have read and approved the final manuscript.

**Financial support.** Anhui University Collaborative Innovation Project of General History of Taiwan Ethnic Minorities (GXXT-2021-034).

**Competing interest.** All authors declare no conflict of interest.

**Ethical statement.** Ethical statement has been important during the entire implementation of this study. The study has been carried out in accordance with the ethical principles of the Institute Review Board (IRB) at the Anhui University of China. This research has obtained ethical approval from the Anhui University Collaborative Innovation Project of General History of Taiwan Ethnic Minorities (GXXT-2021-034).

**Informed consent.** Informed consent was obtained from all individual participants included in the study.

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
