## [Editor Report]

The team of reviewers and myself have been very satisfied with the article, we only have some minor observations that we ask you to clarify. We are sending you the comments below:

Reviewer 1: Accept

The article is fascinating and presents unique perspectives on the subject of investigation.

1. The abstract in this article requires significant modification to meet the standards of an expert audience.

2. Certain sections of the narrative writing lack sufficient citations and need further support.

3. The research lacks a clear aim or objective, which needs to be clarified.

4. The authors need to thoroughly modify the methodology sections, including results and analysis, to enhance the credibility of the study.

5. The study lacks sufficient research on relevant psychological trauma, which needs to be addressed.

6. What are the limitations of the study? This needs to be clearly stated.

7. The implications for future research need to be discussed to provide direction for further studies.

Reviewer 2: Minor revision

The author(s) investigated the impact of climate change on mental health: risks, impacts, and priority actions. I have attached some minor suggestions to the author(s). The findings are interesting, and there is quite a lot to learn from them.

1.The abstract is well written however, it is important to highlight the tool of collecting data questionnaires in this case.

2.In the introduction research gap and how current study fill existing gap should be elaborate clearer. In the last paragraph of the introduction part the object of the study should be elaborate clearer. Please do assign a specific theoretical lens to the study it will make your study more attractive.

3.What sampling method is used? Details about instrument are required. A brief background information of the participants is required. Elaborate the what sampling method is used? Details about instrument are required.

4.In discussion please be focused and compare the results of the current study with existence literature, its compatible or incompatible.

5.Please mention strength and limitation of the study more clearly.

6.The conclusion should be longer with some suggestions for further studies or implications.

7.These will will enhance the manuscript’s clarity and strengthen its overall quality.

Reviewer 3: Minor revision

Thank you for having me review this manuscript. The work is in an important public health area and the manuscript is quite well written. It adds to the growing knowledge of mental health. A few comments to improve the manuscript here below:

1. The way of abstract writing is not scientific. Revise the abstract and write precisely.

2. The introduction section is too short and irrelevant. Authors have missed a lot of important research literature related to the concerned domain. Revise the whole introduction section by using sub-headings.

3. The introductory part is too redundant, should some of the content belong to the literature review section?

4. The conclusion section doesn’t reflect the main part of the study findings. Rewrite it again.

5. The conclusion should be longer with some suggestions for further studies or implications.

6. Sample size and sampling methods need clarification.

Reviewer 4: Minor revision

The authors use 3 rating scales and demographic analysis to analyse how climate change interacts with the population of Pakistani children and adolescents.

Stress variables, climate anxiety, generalised anxiety and depression were assessed. The study shows how these indicators move after heat waves, floods, etc., producing a response in terms of compromising the mental health of children, hypothesising that this compromises the sense of security and growth and the development of the affective and cognitive world at these early stages of neurodevelopment.

It is clear that we need to better distinguish mental health problems from a physiological state of alarm. Furthermore, the alarm is being poorly managed, not just in Pakistan and not just in schools, but by many administrations around the world.

The article comes in the wake of a growing bibliography (page 17 13-15) that underlines how climate change needs to be seen as a challenge and a priority for mental health professionals.

Correctly growing bibliography, underlines how climate change in some respects affects girls more tham males (gender differences), more those who have no acess to resources, belong to minorities or simply thay are poor (socio-economic status).

I would like to explain better what is meant by “Climate change is not only an environmental but also a also a psychological problem” (page 20 22-24). The sentence does not make much sense, it is obvious or it is not clear what it means (I would remove the sentence completely).

I have a more specific final comment: this article focuses mainly on the effects of Extreme Climate Events (ECEs) (pages 6-24-43). It does not consider the ongoing environmental changes that will, unfortunately, be permanent. We don’t know how this will develop in future generations. Permanent changes (long-term climate changes) influence both the evolution of society and the evolution of consciousness and how it will have to adapt.

I recommend mentioning this in the article. This serves to emphasise that the article is aimed at children who can still compare a better before with a worse after (the system is getting worse) and register this as a change in mental health parameters.

In the future, with the collapse of ecosystems, all this may be less possible and everything will turn towards new forms of adaptation, also visible in global and collective terms (Cianconi, P.; Hanife, B.; Grillo, F.; Zhang, K.; Janiri, L. Human Responses and Adaptation in a Changing Climate: A Framework Integrating Biological, Psychological, and Behavioural Aspects. Life 2021, 11, 895. https://doi.org/10.3390/life11090895) The rating scales we use today for climate anxiety may all need to be revised in the face of a permanently hostile environment.

I hope this helps. Thanks to the authors.